# Knowledge Absorption Capacity as a Factor for Increasing Logistics 4.0 Maturity

**Agnieszka Stachowiak [1], Michał Adamczak [2],\*, Lukasz Hadas [1], Roman Domański [1] and Piotr Cyplik [1]**

[1] Poznan University of Technology, Faculty of Engineering Management, 11 Strzelecka Str, 60-965 Poznan, Poland; agnieszka.stachowiak@put.poznan.pl (A.S.); lukasz.hadas@put.poznan.pl (L.H.); roman.domanski@put.poznan.pl (R.D.); piotr.cyplik@put.poznan.pl (P.C.)

[2] Poznan School of Logistics, 6 Estkowskiego Str, 61-755 Poznan, Poland

\* Correspondence: michal.adamczak@wsl.com.pl; Tel.: +48-60-5584-362

**Abstract:** This research strives to show the importance of knowledge absorptive capacity as one of the most important determinants of successful implementation of contemporary solutions and, consequently, development of a company. In the approach presented, the development leads to excellence and is expressed with subsequent maturity levels. The research is focused on identification of the level of absorption of knowledge of contemporary solutions in logistics, grouped in a concept of Logistics 4.0, and how that upgrades the organizational maturity of a company. The research was conducted with CAWI (Computer-Assisted Web Interview), including three questions and a basic query on experts' qualifications. The general conclusion from the research was that to reach a higher level of maturity, a higher level of knowledge absorption is required. However, searching for differences in absorption of solutions within physical flows, information flows and managerial methods seem to be an interesting issue and promising field for further research.

**Keywords:** maturity; knowledge absorption; Logistics 4.0

## 1. Introduction

Contemporary companies can benefit from many solutions, starting from organizational ones, through advanced technical and technological solutions implemented to improve the efficiency of manufacturing processes, to sophisticated methods and tools supporting information processing and flow. On one hand, it is a huge opportunity; on the other, a challenge, as companies need to recognize what the market offers, what they need and can use, and finally to adapt the solutions to use them efficiently. This research strives to show the importance of knowledge absorptive capacity as one of the most important determinants of successful implementation of contemporary solutions and, consequently, development of a company. In the approach presented, the development leads to excellence and is expressed with subsequent maturity levels. The aim of the research is to identify the relationship between the level of knowledge absorption capacity in the organization and the ability to achieve particular maturity levels in the use of Logistics 4.0 solutions in the organization. The research approach to the Logistics 4.0 organizational maturity is based on the definition of logistics where logistics is defined as flow management. The authors then decomposed the definition into material flow management, information flow management, and management methods. Hence, the Logistics 4.0 maturity is analyzed in the context of material flow, information flow, and management methods. The research was addressed to experts (academics and professionals) in supply chain management and logistics, and the results serve as the basis for developing a Logistics 4.0 maturity model based on absorption of solutions that constitute the Logistics 4.0 concept. The paper is structured into four parts.

The first part presents a theoretical background in the following aspects: Logistics 4.0, an organizational maturity model, and knowledge absorption. The second part includes the presentation of the research problem and research methodology, the third discusses the research results, and the fourth is the summary and conclusions presentation.

## 2. Theoretical Background

### 2.1. Logistics 4.0

Logistics 4.0 is a set of technical and organizational solutions designed to improve material and information flows and adjust them to meet the requirements of Industry 4.0 solutions. Logistics are meant to support manufacturing processes; as manufacturing processes shift towards intelligent, autonomous solutions, logistics should follow that shift [1]. Therefore, Logistics 4.0 is inspired and based on the concept of Industry 4.0 [2].

Logistics 4.0 definitions are vague as the concept is not homogenous, being at its initial stage of development [3]. Publications on the subject aim to present Logistics 4.0 as a trend important in logistics [4], solutions within it [5], the role of lifelong learning [6], key competences development [7], and knowledge management. They also refer to the challenges of contemporary markets that logistics have to face, such as information exchange, automation [8], and real-time big data analysis and link Logistics 4.0 to contemporary management paradigms such as sustainability [9]. Additionally, analysis of the literature on the subject brings forth further questions such as which paradigm changes will emerge from the fourth industrial revolution and how to address them proactively together with a balanced review of the variety of approaches considered among professionals in the field of Physical Internet. As a result, the goal for researchers and practitioners is to identify the biggest challenges (technological, societal, business paradigm) of proposed new logistics paradigms as a practical solution in supporting Industry 4.0 emerges [10]. The literature gives some examples of implementation of Logistics 4.0 solutions, elucidates some key aspects of ongoing development and conveys a view of Logistics 4.0, giving examples of some basic technical components. Technological innovation and requirements of modern production are introduced as determinants of the transformation of logistics [11]. Accompanied by decision and information systems with an increasing degree of autonomy, new challenges arise in modelling and implementing autonomous logistics, i.e., Logistics 4.0 [12]. Thus, sophisticated simulation approaches are required, capable of representing both material flow and automation systems as well as autonomous software systems and human actors [13].

Although the systems have autonomous status, they still need to communicate with each other. Communication technologies (i.e., 5G network) play a pivotal role as the key tool that enables the interconnection among isolated business processes, making the most of them and resulting in more powerful business-level applications and decision-making [14]. In Industry 4.0, manufacturing systems go beyond simple connection to also communicate, analyze, and use collected information to drive further intelligent actions. 5G promises to be a key enabler for Factories of the Future, providing a unified communication platform needed to interact with new business models and to overcome the shortcomings of current communication technologies [15]. For 5G mobile communication, five requirements are considered: high data rate and reliability, low latency, increased capacity, increased number of devices, and long battery life. Some of these five requirements are closely related to each other (capacity and the number of devices), and some have a trade-off relationship between each other (reliability and latency) [16].

From an operational perspective, the Logistics 4.0 status is presented in the reports by research centers and logistic services providers [17].

Hence, the list of methods and tools constituting Logistics 4.0 includes sources of data such as smart low battery consuming sensors, GPS, RFID tags, as well as the Internet of Things, drones, and innovative applications, making logistic processes smarter, more connected, automated and

robotized, which undoubtedly improves logistic system performance and contributes to improved performance of supply chains.

The new evolution of the production and industrial process called Industry 4.0, and its related technologies, still have an unknown potential impact on sustainability and the environment. The sustainability impact and challenges of Industry 4.0 could be discussed from four different scenarios: deployment, operation and technologies, integration and compliance with sustainable development goals, and long-run scenarios. Only through integrating Industry 4.0 with sustainable development goals in an eco-innovation platform can it really ensure environmental performance [18]. Strategic, operational, as well as environmental and social opportunities, are positive drivers of Industry 4.0 implementation, whereas challenges with regard to competitiveness and future viability as well as organizational and production fit impede its progress [19]. Industry 4.0 initiatives can help industries to incorporate environmental protection and control initiatives as well as process safety measures in supply chains towards sustainable supply chains. Organizational challenges hold the highest importance followed by technological challenges, strategic challenges, and legal and ethical issues [20].

### 2.2. Organizational Maturity

Maturity can be defined as "the state of being complete, perfect or ready" [21]. Maturity is referred to as the state of growth, level of excellence, as in [22], where the process of bringing something to maturity means bringing it to a state of full growth, and to improvement and excellence [23].

Naturally, the term "maturity" is used in psychology and other human sciences, but it is also used in management in the context of quality [24]. Since the 1970s, maturity models have been recognized as an important concept for the improvement of organizations. Crosby was among the first to propose, in 1979, a quality management model with five levels of maturity [25]. To set the guidelines or references to the development or perfection, maturity models have been developed.

Maturity models are now widely spread in Project Management (PM), Knowledge Management (KM), Information Systems (IS) and Supply Chain Management (SCM) industries [26]; more than one hundred maturity models have been published as guidance towards business excellence, as seen in Figure 1.

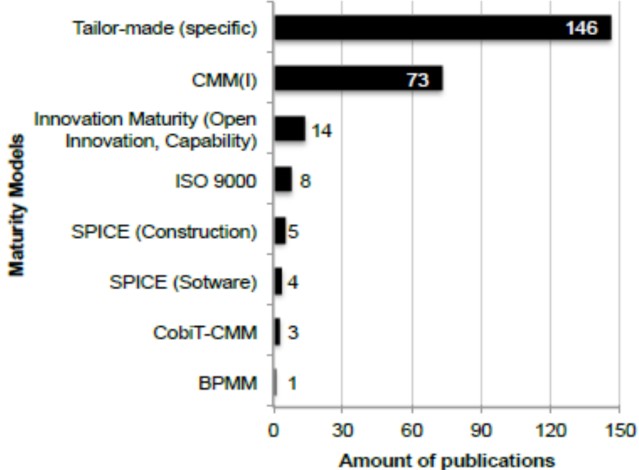

**Figure 1.** Published maturity models by area [26].

Maturity models in literature have different characteristics and structure; they can be of moderate or high complexity, and maturity levels can be described in simple or complex terms.

Probably the most disseminated maturity model is the Capability Maturity Model (CMM) developed by researchers at the Software Engineering Institute of the Carnegie Mellon University [24].

In the field of Industry 4.0, many researchers have developed their own maturity models. For example, Schumacher et al. [27] developed a model for assessing Industry 4.0 maturity with nine

dimensions defined and 62 characteristics assigned to them. The dimensions: "Products", "Customers", "Operations", and "Technology" have been determined to assess the basic enablers. Additionally, the dimensions "Strategy", "Leadership", "Governance", "Culture", and "People" have been depicted for including organizational aspects into the assessment.

Each maturity model is based on the assumption that the maturity or quality of a process or project directly correlates with its success [28] as the term maturity is directly related to excellence [29]. Achieving excellence is a process that requires taking specific initiatives. To facilitate the process and organize the way to excellence, maturity levels are distinguished. Completing certain requirements means reaching the predefined level of maturity. Various maturity models are structured differently, with four up to seven levels, with the lowest level representing lack of signs of the approach the maturity is referred to, while the highest level represents full implementation of the approach; its deep understanding.

## 2.3. Knowledge Absorption

To grow, develop, and be innovative, companies need knowledge. They can create it themselves or acquire it from the environment. The knowledge acquired should be used to create competitive advantage; thus its absorption is a crucial process. Hence, absorption capacity (also called diffusion) is considered a measure of the implementation of technological solutions. The absorption capacity most often concerns the implementation of new production processes, the use of new technologies, learning, etc.

A company with higher absorptive capabilities possesses better learning abilities and foresees opportunities beyond its horizon [30]. Therefore, the concept of absorption capacity is considered one of the most important concepts to have emerged in the field of organizational research in recent years.

There are many different definitions of absorption capacity in the relevant literature. In the initial studies, the absorption capacity of a company was treated in one-dimensional terms. Nowadays, researchers agree that this concept is multidimensional, although there is still no consensus as to how many dimensions it covers and what their specificity is [31].

Nevertheless, as the complexity of the concept is broadly agreed on [32], many theoreticians in this area structure absorptive capacity dividing it into dimensions and components [33] where dimensions include acquisition, assimilation, transformation and exploitation of knowledge [34], as listed and defined below [35].

I.   Ability to acquire knowledge (Aac): understood as the ability to locate and identify sources, and also its valuation and acquisition as a basis for operational activities.
II.  Ability to assimilate knowledge (Aas): understood as the ability to assimilate it, consisting of ordering, correct analysis, interpretation, and understanding.
III. Ability to transform knowledge (Atr): understood as the ability to combine new knowledge with knowledge already possessed, through adding or eliminating specific components of knowledge or by interpreting knowledge in a new innovative way.
IV.  Ability to use knowledge (Aus): understood as the ability to incorporate knowledge into operational activities and improve processes and competences.

Based on the structure of the concept, knowledge absorption requires an active approach, openness to interaction with business and scientific environments, and knowledge flows and diffusion within the company itself. Moreover, the findings from the study by Kostopoulos et al. [35] show that firms' involvement in innovation collaborations with various external parties enriches their knowledge base [36] and develops a better ability to assimilate and exploit external knowledge [37], to facilitate the understanding of new concepts and effectively implement innovation [38].

For the purposes of analyzing the absorption capacity, conceptual models of the conditions, components, and results of the absorption of knowledge, along with the relations between them, are very useful.

## 3. Research Framework

### 3.1. Research Methodology

The research was conducted to identify the knowledge absorption capacity level enabling an upgrade to a higher maturity level. The organizational maturity model that was used as the basis for the research questions and goals formulation was composed of five levels (Ignoring, Defining, Adapting, Managing, and Integrated, based on the approach presented in ISO standards) and referred to three areas: management, material flow, and information flow (Table 1). Material (physical) and information flow, together with financial flow are basic components of business, thus their analysis and assessment seems to be important for management in all its perspectives and dimensions. They are widely used in performance models, i.e., GSCF (Global Supply Chain Forum), SASC (Strategic Audit Supply Chain), WCL (World Class Logistics), ALSOG (Association Française pour Logistique), AFNOR (Association Française de Normalisation), SCM/SME (Supply Chain Management/Small and Medium Enterprise), SCOR (Supply Chain Operation Reference-Model), APICS (American Production and Inventory Control Society), ECR (Efficient Customer Response), EFQM (Excellence model by European Foundation for Quality Management) [39], hence the flow logic is widely accepted. This is why the authors decided to consider material and information flow in their maturity model. Nevertheless, the financial flow was replaced with a management aspect, as logistics solutions are generally not dedicated for financial flow, whereas in many cases they are of a wider range than physical or information flows striving for integration and synergy. Moreover, management methods cover the important aspects of business strategic management, human resources management, and others; integrating them with material and information flows into corporate management. For these cases, the management aspect was selected as one of the maturity model dimensions. The model refers to the organization and the solutions are within Logistics 4.0.

**Table 1.** Logistics 4.0 maturity model (based on [29]).

| Aspect | Ignoring | Defining | Adopting | Managing | Integrated |
|---|---|---|---|---|---|
| **Management** | not aware of the need for integration | see the need for integration but do not know how to manage it | integration is initiated | integration at most levels | full integration resulting in synergy |
| **Material Flow** | do not know about advanced solutions improving material flows | know about advanced solutions improving material flows but do not use them | some advanced solutions improving material flows are implemented | many advanced solutions improving material flows are implemented | all possible advanced solutions improving material flows are implemented |
| **Information flow** | do not know about advanced solutions improving information flows | know about advanced solutions improving information flows but do not use them | some advanced solutions improving information flows are implemented | many advanced solutions improving information flows are implemented | all possible advanced solutions improving information flows are implemented |

The assessment of maturity level is based on analysis of Logistics 4.0 dimensions. The aspects to be assessed include the need and the level of integration of internal processes, and, when applicable, the supply chain, and number and scope of advanced solutions improving material and information flows.

In the research, the authors focused not on a particular level but on transitions between maturity levels of Logistics 4.0. The absorption capacity was described in four levels (a detailed description of the levels is provided in Section 2.3.). Characterizing the knowledge absorption capacity in an

organization by defining its levels is synonymous with the fact that reaching a given level requires prior achievement of all lower levels. This fact was included in the research, in which the respondents determined the required level of knowledge absorption capability for the use of Logistics 4.0 solutions in three distinguished aspects of the logistics system:

- material flow: it is understood as the use of technical and organizational solutions, aimed mainly at increasing the efficiency of the flow of material (raw materials, components, final goods: in the area of supply, production, warehouse, and distribution networks),
- information flow: it is understood as the use of IT and organizational solutions aimed mainly at increasing the availability, timeliness, and correctness of information used for the efficient and effective implementation of logistics processes,
- management: all organizational and IT solutions are adopted to improve the functioning of the logistics system (its efficiency), including cooperation with suppliers, partners in the supply chain, and distribution networks, as well as internal logistics processes.

The respondents' group included researchers dealing with Logistics 4.0, business practitioners, and people operating in both domains (researchers—61%, practitioners—17%, people operating in both domains—22%). Experts in the self-assessment questionnaire assessed their knowledge of the Logistics 4.0 issues on a five-point scale (where five is the highest level of knowledge). Answers of respondents who assessed their knowledge at a level lower than four were rejected as unreliable. As a result, responses were received from 18 respondents (83% of them identified their level of knowledge of Logistics 4.0 as four and 17% as five) which was the basis for analysis.

*3.2. Research Goal*

The research was designed to investigate (1) whether there is any connection between knowledge absorption (concerning Logistics 4.0 solutions) and organizational maturity level, and (2) what are the determinants or minimal requirements for a company to upgrade its maturity.

The goal of the research conducted on the group of experts was to identify the relationship between the level of knowledge absorption capacity in the organization and the ability to achieve particular maturity levels in the use of Logistics 4.0 solutions in an organization. Respondents were asked to determine what the necessary level of knowledge absorption was to allow the organization to upgrade through successive maturity levels of Logistics 4.0.

*3.3. Research Tool*

The research was conducted using CAWI (Computer-Assisted Web Interview) including three questions and a basic query on experts' qualifications. The first question was on material (physical) flow. The respondents were asked what the required level of knowledge absorption capacity is in the organization in order to upgrade the maturity level. The question referred to knowledge on Logistics 4.0 solutions supporting material flow absorption. The second question was on information flow, and the respondents were again asked what the required level of knowledge absorption capacity is in the organization in order to upgrade the maturity level. However, this time the question referred to knowledge on Logistics 4.0 solutions supporting information flow absorption. The third question was on management, and the respondents were consequently asked what the required level of knowledge absorption capacity is in the organization in order to upgrade the maturity level. This time, the question referred to knowledge on Logistics 4.0 solutions supporting management. Each aspect was explained with a brief introduction.

## 4. Research Results Discussion

The level of the knowledge absorption capacity required for the transition between successive maturity levels of Logistics 4.0 was determined based on analysis of the answers of experts. Analysis

of the results shows the following in all areas: flow of materials (Figure 2), information flow (Figure 3) and management (Figure 4):

- "Transition from Ignoring to Defining" requires having "Ability to acquire knowledge".
- "Transition from Defining to Adopting" requires having "Ability to assimilate knowledge".
- "Transition from Adopting to Managing" requires having "Ability to transform knowledge".
- "Transition from Managing to Integrated" requires having "Ability to use knowledge".

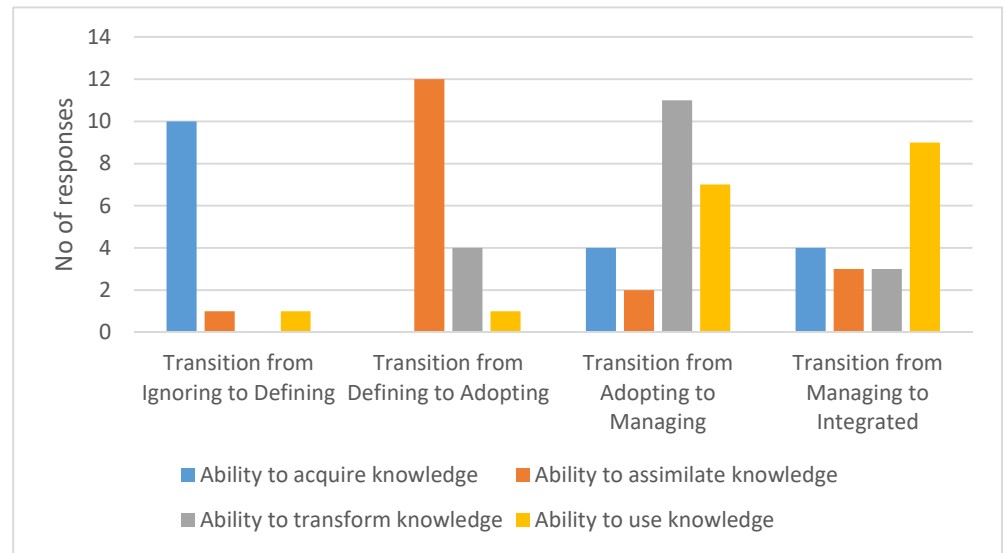

**Figure 2.** Research results in material flow area (own study).

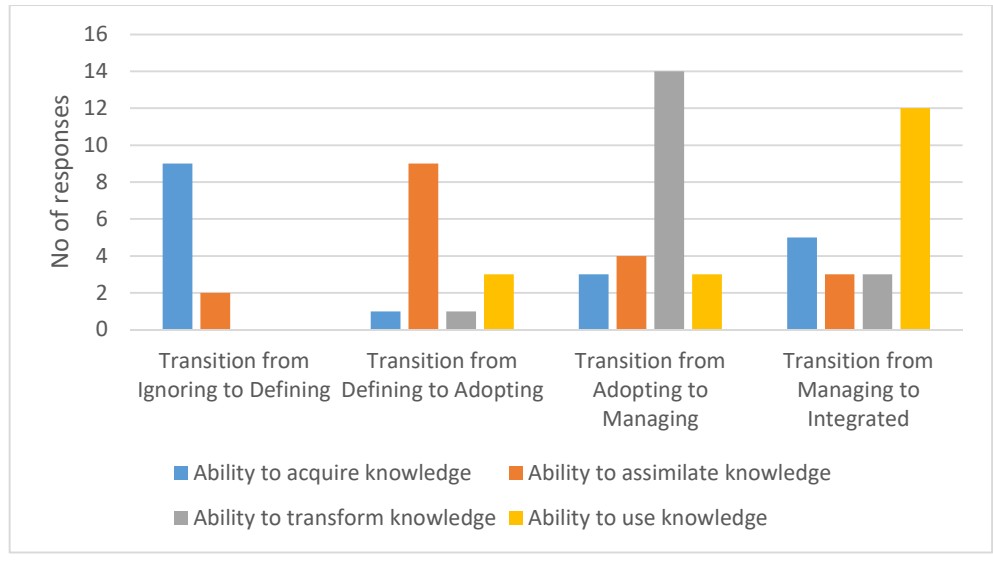

**Figure 3.** Research results in information flow area (own study).

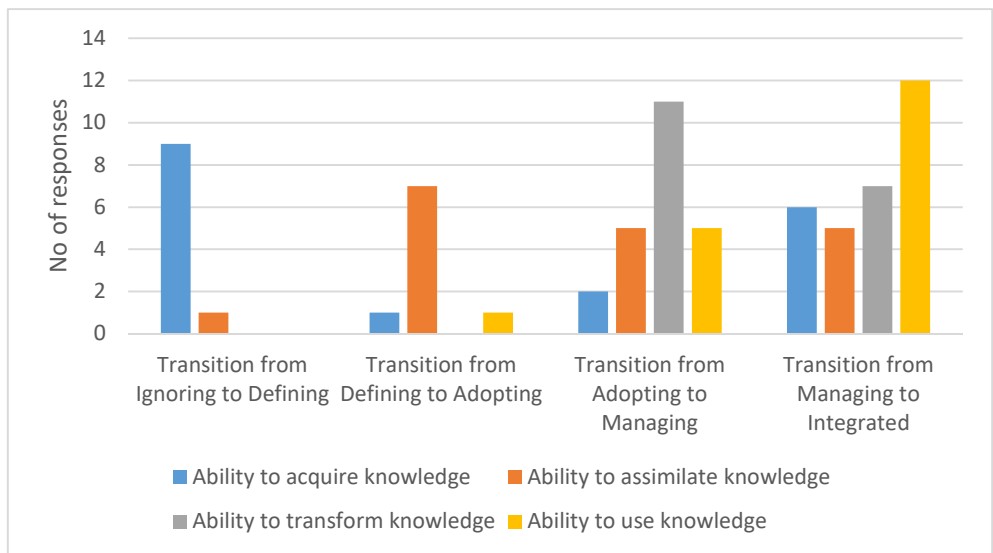

**Figure 4.** Research results in management area (own study).

The diagrams with research results present the relationship between the level of knowledge absorption capacity in the organization and the ability to achieve a certain Logistics 4.0 maturity level. The dependence is directly proportional, i.e., the achievement of higher Logistics 4.0 maturity levels in an organization requires achieving ever-higher levels of knowledge absorption capacity in the organization.

Detailed analysis of the results allows noting the differences between knowledge absorption capacities in the three areas of the logistic system: material flow, information flow, and management. In the area of material flow, the logistics system general requirements to the level of knowledge absorption capacity as a factor limiting the achievement of a certain Logistics 4.0 maturity level are the lowest. These requirements grow for the information flow area to reach the highest level for the management area. To sum up, the research indicates that management is the most demanding area of the logistics system. As an element based on knowledge processing, it requires the greatest absorption capacity to implement Logistics 4.0 solutions.

## 5. Conclusions

The ability to absorb knowledge in an organization is a key element in implementing innovative solutions in material, information, and management flows. The ability to use knowledge gives the greatest possibility of using solutions falling within the scope of Logistics 4.0. Logistics 4.0 is a set of solutions striving to improve companies' performance and support Industry 4.0 concept implementation. These solutions are highly advanced and their implementation requires extensive knowledge and understanding. Thus, the implementation of contemporary solutions in the field of Logistics 4.0 requires that the organization first improve the absorption of knowledge. This ability is the ground on which an effective logistics system can be built using modern technical, technological, and organizational solutions.

The research conducted was intended to be a pilot study to bring inspiration and structure for further research areas. Since the problem analyzed is broad and multi-dimensional, it requires deepened analysis. The authors intend to focus their further efforts on interdependencies between the dimensions of the maturity model (material flow, information flow, and management) in the context of the knowledge absorption level required to upgrade their maturity levels.

Importantly, the research results confirmed that experts see the relationship between knowledge absorption levels and Logistics 4.0 maturity levels. Therefore, it is legitimate to conclude that both models can be used complementarily and should form the basis for planning activities under the

implementation of detailed solutions (regarding changes in management, organization of material flows, and information flow). The identified dependency should also be taken into account when analyzing a company's strategic readiness for implementing the concept of Logistics 4.0, especially in the aspect of human resource management.

The topic presented in the research is becoming more and more up-to-date due to the development of the 5G network, which gives the opportunity to create a widely, and publicly, available communication infrastructure for solutions included in Logistics 4.0. Thus, thanks to the development of 5G technology, the costs of implementing and using Logistics 4.0 solutions are reduced, which will probably contribute to their dissemination. Identifying the factors of the effective use of Logistics 4.0 tools resulting from the ability to absorb knowledge in an organization is becoming an important topic from the point of view of building sustainable enterprises and sustainable supply chains.

The general conclusion from the research was that to reach a higher level of maturity, a higher level of knowledge absorption is required. However, searching for differences in absorption of solutions within material flows, information flows, and managerial methods seems to be an interesting issue and promising field for further research.

**Author Contributions:** Conceptualization, A.S. and L.H.; methodology, L.H. and M.A.; formal analysis, P.C.; investigation, M.A.; resources, A.S. and R.D.; writing, original draft preparation, M.A.; writing, review and editing, A.S.; supervision, L.H.; funding acquisition, L.H.

**Funding:** This research was funded by the Poznan University of Technology, Faculty of Engineering Management, Project Number 11/140/SBAD/4168.

**Conflicts of Interest:** The authors declare no conflicts of interest. The funders had no role in the design of the study; in the collection, analyses, or interpretation of data; in the writing of the manuscript, or in the decision to publish the results.

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
