# Peer review of "Knowledge Absorption Capacity as a Factor for Increasing Logistics 4.0 Maturity"

_applsci, doi:10.3390/app9245365_

Round 1

Reviewer 1 Report

This is an interesting submission, however I believe the following amendments would be required to make it publishable:

Some of the key sources on absorptive capacity could be cited, and overall you should include more up to date sources, The figures should be redrawn; the first figure is blur, and the charts are missing the data labels, The research results would require expansion; it is not clear how you reached the conclusions, I believe some of the arguments in your results and discussion sections could be cross referred to literature to substantiate your arguments.

Thank you.

Author Response

In the paper more up-to-date sources are used. Old books and papers are replaced by newer edition or by newer results of studies.

Figure 2 is redrawn. Figure 1 is in better quality.

On chart missing data labels are added.

In the conclusions deeper explanation is added.

Reviewer 2 Report

Line 21: CAWI ? abbrevation not explained

Line 22: a higher level .. (add ‘a’)

Line 23: a higher level (add ‘a’)

Line 23: required. However …

Line 36: presented, the development … (add comma)

Line 39: in the organization (add ‘the’)

Line 39: In the ability (add ‘the’)

Line 40: Research à research

Line 43: are to be the basis à serve as the basis

Line 44: developing a Logistics .. (add ‘a’)

Line 45: is presentation of à presents a

Line 46: organizational maturity model

Line 46: knowledge absorption

Line 46: parts à part

Line 47: is discussion on à discusses the ..

Line 54: drop ‘and’

Line 56: at its initial stage

Line 59: management. (add ‘.’)

Line 72: both : à both (drop ‘:’)

Line 74: the Logistics 4.0 status

Line 92: Knowledge Management (KM)

Line 105: process [20]. (add ‘.’)

Line 108: drop ‘A.’

Line 115: drop ‘a’

Line 118: 4,5, or event 7 levels à 4 up to 7 levels

Line 123: absorption à ‘absorption’

Line 124: subject à relevant

Line 125: the company à a company

Line 128: ‘absorption capacity’ appears for the second time between (..). No need ?

Line 135: considered to be the one

Line 138: drop ‘R.’

Line 150: level à levels

Line 155: I think (Aat) is not the right abbreviation

Line 158: (Aas) appears for the second time (see line 153). Correct this.

Line 162: The Zahra and George model

Line 165: is à are

Line 175: that is used

Line 178: economy à business

Figure 2: why has the third row of the figure a dark background? It is less readable, please make it white background

Line 195: In research, (add comma)

Line 195: levels à level

Line 200: 3 à three

Line 219: see à investigate

Line 219: is à exists

Line 219: knowledge à knowledge absorption

Line 228: ‘CAW’I: no explanation of the abbreviation

Line 240: what is ‘mode analysis’? Is it common knowledge, I do not think so.

Line 260: achieve a certain .. (add ‘a’)

Lines 264 till 271: this is no real discussion of the results. There should be more explanation, separate for Figure 3, 4 and 5 so the reader can understand, step by step, your conclusion. So at least add another paragraph explaining this.

Line 273: structure for (add ‘for’)

Line 276: of the maturity model

Line 279: required. However ..

Line 279: physical à material

Line 350: drop ‘(‘

Line 355: ‘Procedia CIRP’ italic

Line 364: ‘Organization Science’ italic

Line 368-369: ‘Academy of Management Review’ italic

Line 380: ‘Strategic’ italic

Author Response

All spelling mistakes are corrected.

Reviewer 3 Report

Dear authors,

I started the review of your article with enthusiasm because of the very promising title, but unfortunately, my enthusiasm went away very soon.

First of all, after performing the plagiarism check (with Turnitin) I found out quite a high similarity index (32%). Soon I discovered the chapter published in Intelligent Systems in Production Engineering and Maintainance (Springer Nature, January 2019) and the parts of the text and Figure 2 that are identical, but they are either not referenced at all in your article either are referenced with not all equal references. I couldn't find the mentioned chapter, which was a "preliminary communication" of some research results described by this article, among the references of your article.

There are other "problems" present in this article: the language is very poor - there are many many mistakes. Sometimes is hard for reading. The research methodology is very poorly described (it doesn't allow to repeat your research what is the precondition for scientific article), the presentation and the discussion of results are also very poor. In too many places the references are missing or they are incorrectly placed within the sentence/paragraph. Figure 2 is a result of your previous study, but it was already published (by different authors that also claim that is a result of their own study). In lines 177 and 178, you start to use the terms "material, financial and information flow" but the definitions only follow in lines 202-210. Through the article, many abbreviations are used that are not all explained. 

The articles need to be revised through roughly to be published with the journal as Applied Sciences, but, unfortunately, at the moment, I have to decline the publication. 

Author Response

All elements which are included in another publication Springer Nature, January 2019 are changed or references are added.

Text is proofread, spelling and gramar mistakes are improved.

Disscussion of a results in conlusion part is extended. 

References are added and improved plaicing of references in sentences and in paragraphs.

Reference to figure 2 is added.

In line 202-210 (paper before revision) are description of flows in Logistics 4.0 area. In previous lines material, informacion and financial flowas are presented as dimensions described by another authors.

Abbreviations are described and explain as soon as possible.

Round 2

Reviewer 1 Report

Thank you for the efforts in revising the manuscript. It is clear that the authors have taken into account the original comments that were provided. I am happy with the  changes and the authors' response.

Author Response

We will try to make a next proofreading to improve an English language in the paper.

Reviewer 3 Report

Dear authors,

I see that you followed majority of my comments and suggestions regarding the referencing and including missing references in the Reference list, but there are still some improvements needed: you should avoid the referencing as you have in lines 76, 90, 111, 117, 162, 177, etc. and I still think not all sources were included.

Author Response

Referencing in lines 76, 90, 111, 117, 162, 177 are corrected. All references are checked one more time.
Next proofreading was made. Some grammar and spelling corrections are made.